# Robust Domain Generalisation with Causal Invariant Bayesian Neural Networks

## Abstract

Deep neural networks can obtain impressive performance on various tasks under the assumption that their training domain is identical to their target domain. Performance can drop dramatically when this assumption does not hold. One explanation for this discrepancy is the presence of spurious domain-specific correlations in the training data that the network exploits. Causal mechanisms, in the other hand, can be made invariant under distribution changes as they allow disentangling the factors of distribution underlying the data generation. Yet, learning causal mechanisms to improve out-of-distribution generalisation remains an under-explored area. We propose a Bayesian neural architecture that disentangles the learning of the the data distribution from the inference process mechanisms. We show theoretically and experimentally that our model approximates reasoning under causal interventions. We demonstrate the performance of our method, outperforming point estimate-counterparts, on out-of-distribution image recognition tasks where the data distribution acts as strong adversarial confounders.

## 1 Introduction

The training of deep neural networks commonly relies on the assumption that the distribution of the training data is representative of the distribution at inference. Despite being widely adopted, this assumption has been heavily criticised as it is often challenged in practice (Langford, 2005; Ben-David et al., 2006; Albuquerque et al., 2019; Arjovsky et al., 2019; Jalaldoust & Bareinboim, 2024). Indeed, despite tremendous progress on many vision tasks over recent years, deep neural networks face limitations and reduced performance on tasks requiring the model to shift from its training distribution at test time (Mao et al., 2022).

Causality theory under the do-calculus framework (Huang & Valtorta, 2006; Pearl, 2009) provides tools to explain these limitations. Neural networks extract correlation patterns but do not possess knowledge on the cause-effect relationships from the data; causal links and potentially spurious correlations are learned alike. However, the two relationships fundamentally differ as correlations can be non-causal: e.g. a correlation between $X$ and $Y$ can be explained by a causal link $X \leftarrow Z \rightarrow Y$. This type of correlation can be specific to the distribution if $Y$ depends on the domain, e.g. *has tusks and a prehensile trunk $\leftarrow$ African bush elephant $\rightarrow$ is in a savannah environment*. A system only learning the correlation between $X$ and $Y$ may fail to recognise an elephant in a different environment. However, direct causal links make these relationships explicit and are therefore more robust to changes in the distribution (Schölkopf et al., 2021). In particular, the *Independent Causal Mechanisms* principle states that causal relationships are only sparsely connected and altering one should not modify the others (Peters et al., 2017; Schölkopf et al., 2021), allowing some learned mechanisms to be *transported* to new environments. Indeed, changes in the distribution can be modelled via the notion of *transportability* of causal mechanisms (Bareinboim & Pearl, 2013; Jalaldoust & Bareinboim, 2024). As such, transportable causal effects are equivalent to domain-invariant features. These principles motivate the search for factorised neural models composed of independent modules representing either domain-specific or domain-invariant conditional distributions.

Bayesian neural networks (BNNs) (MacKay, 1992; Blundell et al., 2015b; Magris & Iosifidis, 2023) are another line of work attempting to model robust representations of mechanisms by learning the distribution $W$ of a parametric function $y = f_w(x), w \sim W$ instead of a single point estimate $w$. BNNs are particular suited to express uncertainty in their response and reduce overconfidence. By

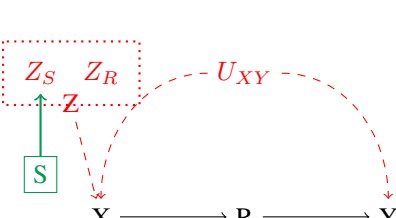

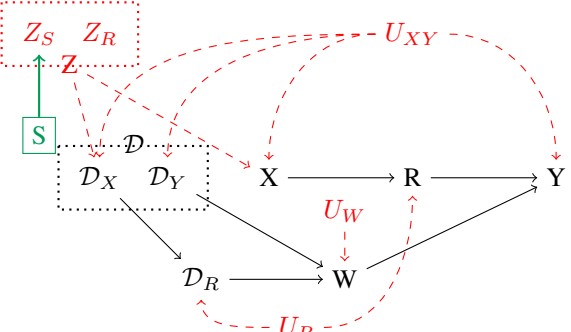

(a) Target causal mechanism. $R$ is a latent representation of $X$ estimating $Z_R$. This representation allows computing interventional queries transportable across domains using $R$. However, it is not directly accessible via supervised learning.

(b) Causal graph during training. The inference mechanism determining $Y$ is estimated using a function $f_w(r)$ parametrised by the weights $w \sim W$ and with an input representation $r \sim R$. $W$ is learned from the training data $\mathcal{D}$ (training target $\mathcal{D}_Y$ and representation $\mathcal{D}_R$ learned from $\mathcal{D}_X$) and stochastic processes represented by the random variable $U_W$.

Figure 1: Target causal representation and actual causal graph during training. $X$ is the input image and $Y$ is the output class. $Z$ is the variable representing the factors of distribution generating the input $X$, it is composed of domain-specific factors $Z_S$ and robust domain-invariant factors $Z_R$. $U_{XY}$ represents the shared factors of variations between $X$ and $Y$. The variables in red are unobserved. $S$ is a selection variable determining the current domain. The target causal graph is different from the one used in supervised deep learning as it omits the influence of the datasets $\mathcal{D} = \{\mathcal{D}_X, \mathcal{D}_Y\}$. The datasets depend on the same factors as $X$ and $Y$ and add spurious correlations that are not captured when using the target representation.

differentiating the modelling of uncertainty in the data distribution (aleatoric uncertainty) from the uncertainty in the process distribution (epistemic uncertainty), BNNs can better capture the latter (Kendall & Gal, 2017). However, BNNs are not fundamentally more robust against distribution shifts than their point-estimate counterparts (Izmailov et al., 2021). This problem can be explained by the intractability in the optimisation of the posterior (Magris & Iosifidis, 2023). Motivated by the use of partially stochastic networks (Sharma et al., 2023), we argue that BNNs should also be integrated in a wider framework estimating causal mechanisms.

We propose a Causal-Invariant Bayesian (CIB) neural network taking advantage of the causal structure of the supervised learning process to differentiate domain-specific and domain-invariant mechanisms. Compared to previous work, we integrate a more realistic causal graph that takes into account the Bayesian model update during learning. We use variational inference and partially-stochastic Bayesian neural networks to model the causal paths in an interventional setting. We verify theoretically and experimentally the applicability of our method. In particular, we perform tests on standard out-of-distribution (o.o.d) image recognition tasks. Our contributions are as follows:

- We reformulate the Bayesian inference problem to include causal interventions and the supervised learning mechanism and propose a factorised model explicitly modelling domain-invariant mechanisms in an unsupervised fashion,

- We propose an architecture wrapped around a neural network inspired by this factorised model and test it on challenging tasks requiring domain shifts,

- We show that adding our model can increase the i.i.d and o.o.d performance of the underlying neural network and reduce overfitting,

- Additionally, we find with our model that adding unsupervised contextual information with a Bayesian classifier to a standard ResNet improves its robustness and training stability, even with a small number of context and Bayesian weight samples.

Our code is available at this anonymous repository: `https://anonymous.4open.science/r/cibnn-1B0B`.

## 2 BACKGROUND AND RELATED WORK

**Causal Inference and Transportability**   Causal inference methods aim to estimate the result of causal queries. Answering these queries require an inference model to have a partial knowledge of the causal mechanisms underlying the studied system. The causal knowledge is typically divided into three layers: *observational* (*l1*), *interventional* (*l2*) and *counterfactual* (*l3*) (Pearl, 2009). Bareinboim et al. (2022) showed that these layers form a non-collapsing hierarchy, i.e. information at level $i$ is needed to answer a question at level $i$. Richens & Everitt (2024) further showed that an agent must learn a causal model of the world (at level *l2*) to robustly solve a task, e.g. subject to changes of domain. The problem of formally representing causal quantities is addressed by the do-calculus framework, which introduces the $do(\cdot)$ operator corresponding to an intervention on a causal variable (Pearl, 2009). For example, given a treatment $T$ and background covariates $X$, the probability of getting cured given by $C$ can be expressed with the query $P(C|do(T), X)$. The do-operation implies that the treatment is received in an unbiased manner (i.e. as in a double-blind study). An interventional query as described in the example can be reduced to observations using the rules of do-calculus. We describe them in Appendix A.1. *Transportability* theory furthermore allows representing causal mechanisms across multiple domains (Bareinboim & Pearl, 2013) by differentiating domain-specific and domain-invariant variables in a causal graph. Assuming the domains can be represented with a unified causal graph, domain-specific mechanisms can be modelled by a *selection variable* whose value depends on the current domain. The *S-admissibility* criterion (Jalaldoust & Bareinboim, 2024) states that the value of a causal variable $A$ is invariant when shifting from a domain $\mathcal{M}_i$ to a domain $\mathcal{M}_j$ if it can be *d-separated* from all selection variables $S_{ij}$ (given observations $Z$), i.e. $A \perp\!\!\!\perp S_{ij}|Z \implies P^i(A|Z) = P^j(A|Z)$. Mao et al. (2022) also integrate causal transportability for vision tasks but do not include the Bayesian learning process. They combine input and contextual information differently and only select context with the same label during training. Conversely, we make the assumption that the context should be diverse and representative of the label distribution. We use label mixup to this end. We also include new regularisation techniques.

**Disentanglement**   A large body of work attempting to disentangle domain-specific and domain-invariant information rely on an autoencoding paradigm and reconstruct the input image as part of their training process (Gabbay et al., 2021; Yang et al., 2021; Zhang et al., 2022). For instance, Zhang et al. (2022) build a latent space regularised to divide domain-invariant *semantic* features from a domain-specific *variation* space. Similarly to our work, the authors add contextual information by feeding different samples to a variation space encoder to improve domain generalisation. However, they mainly rely on a reconstruction loss whereas we argue that this component is not needed for building robust representations. van Steenkiste et al. (2019) argued that having disentangled representations improved systematic generalisation in abstract visual reasoning tasks. The authors also found that a low reconstruction error was not necessary for performance on downstream tasks. Our work differs from the disentanglement literature by detaching from the autoencoding paradigm and adopting a causal Bayesian approach.

**Bayesian Neural Networks and Variational Inference**   Bayesian deep learning methods teach neural networks to simulate Bayesian reasoning when learning new information. This approach allows dealing with uncertainty and has been argued to help mitigate overconfidence and improve robustness in neural networks (Magris & Iosifidis, 2023). Bayesian Neural Networks (BNNs) model the distribution of parametric functions solving a task: the parameters $w$ of a BNN are not directly optimised but sampled from a learned posterior distribution $P(w|\mathcal{D})$. This distribution can be obtained via the *Bayes objective*: $P(w|\mathcal{D}) = \frac{P(\mathcal{D}|w)P(w)}{P(\mathcal{D})}$. However, training BNNs is challenging because the computation of the denominator (called the *evidence*) is often intractable (Izmailov et al., 2021). A standard way to circumvent this issue is to approximate the posterior $P(w|\mathcal{D})$ with a *variational distribution* $q(w)$. This distribution can be estimated by maximising the *Evidence Lower Bound* (ELBO) (Blundell et al., 2015b). This quantity can have many expressions, the one most suitable for our work is shown in Equation 1. It simultaneously maximises the likelihood of the data $\mathcal{D} = \{\mathcal{D}_x, \mathcal{D}_y\}$ given the model $w$ while maintaining the distribution $q(w)$ close to the prior $P(w)$.

$$\text{ELBO}(q) = \mathbb{E}_{q(w)}[\log P(\mathcal{D}_y|\mathcal{D}_x, w)] - \text{KL}(q(w)||P(w)) \tag{1}$$

Izmailov et al. (2021) showed that the posterior distribution of BNNs have a high functional diversity and can outperform their *point-estimate* counterparts on downstream tasks. However, they do not necessarily show a high diversity in the parameters space and strictly following Bayesian posteriors can lead to reduced robustness against distribution shifts. Sharma et al. (2023) further showed that partially-stochastic BNNs outperform networks containing only Bayesian layers. Roy et al. (2024) recently showed that inducing BNNs to assign similar posterior densities to reparametrisations of the same functions (i.e. different weights realising the same function) could improve their ability to fit the data. These findings motivate our work on integrating BNNs as a part of a larger causal neural network.

**Mixup**  Input Mixup (Zhang et al., 2018) and Manifold Mixup (Verma et al., 2019) are regularisation techniques for improving robustness in image classification tasks by interpolating inputs, respectively latent representations. The main principle of mixup consists of combining two inputs $(x_i, x_j)$ and labels $(y_i, y_j)$ together to form a new virtual training example $\overline{x}$ with label $\overline{y}$: $\overline{x} = \alpha \cdot x_i + (1 - \alpha) \cdot x_j$ and $\overline{y} = \alpha \cdot y_i + (1 - \alpha) \cdot y_j$ (with $\alpha \in [0, 1]$ constant). Manifold Mixup performs a similar operation but interpolates latent representations instead of the inputs. Gendron et al. (2023) argued that mixing could be used to include interventional information into the network's training. Following these observations, we integrate mixup strategies as part of our training scheme.

## 3 INTERVENTIONAL BAYESIAN INFERENCE

We propose to learn deep domain-invariant representations that can be leveraged to solve o.o.d tasks. In this section, we show the necessary assumptions needed to learn this representation in a supervised fashion and identify an interventional query that can solve the task. We show that this query can be answered using Bayesian Neural Networks (MacKay, 1992; Blundell et al., 2015b; Magris & Iosifidis, 2023). Section 4 describes our proposed architecture.

**Interventional queries**  The causal graph in Figure 1b shows the causal dependencies between the variables involved in the learning process. The standard inference query $P(Y|x)$ is affected by spurious correlations as they do not distinguish the causal links $X \xrightarrow{\cdots} Y$ (i.e. $X \to \cdots \to Y$) and their common causes $Z \xleftarrow{\cdots} U_{XY} \xrightarrow{\cdots} Y$. In causality theory, the $do(\cdot)$ operator removes the incoming dependencies of the target variable (Pearl, 2009). Therefore, the interventional query $P(Y|do(x))$ solves the confounding issue (Mao et al., 2022). However, it does not take into account the conditioning on the training data $\mathcal{D}$. We specify an interventional query that makes the dependency on the data explicit. Note that we simplify the formalism by considering the label $Y_{\text{true}}$ and predicted value $Y_{\text{pred}}$ with a single variable $Y$. This is equivalent to having a prediction loss $\mathcal{L}(Y_{\text{pred}}, Y_{\text{true}}) = 0$.

Conditional distributions independent from the domain selection variable $S$ are transportable across domains (Jalaldoust & Bareinboim, 2024). This independence can be obtained using the $do(\cdot)$ operation by removing the outgoing dependencies of S. The conditional distribution $P(y|do(x), do(\mathcal{D}_X), \mathcal{D}_Y)$ is transportable across domains but is not tractable as it requires marginalising over all possible input distributions (via the frontdoor criterion). We give more details in Appendix A.3. Instead, we use a partially transportable query $P(y|do(x), \mathcal{D}_X, \mathcal{D}_Y)$. Equation 2 formulates this query using observations only (proof in Appendix A.2):

$$P(y|do(x), \mathcal{D}_X, \mathcal{D}_Y)$$

$$= \underbrace{\int_w P(w|\mathcal{D}_X, \mathcal{D}_Y)}_{\text{marginalisation over W}} \underbrace{\int_r P(r|x, w, \mathcal{D}_X, \mathcal{D}_Y)}_{\text{marginalisation over R}} \underbrace{\int_{x'} P(x'|\mathcal{D}_X, \mathcal{D}_Y)}_{\text{marginalisation over X}} \overbrace{P(y|x', r, w, \mathcal{D}_X, \mathcal{D}_Y)}^{\text{inference}} dx'\, dr\, dw$$

$$(2)$$

The interventional query must be decomposed into four components to be represented using observations only. The marginalisation over W is equivalent to an application of the backdoor criterion and ensures that no backdoor paths exist between $Y$ and $W$. The marginalisations over $X$ and $R$ are applications of the frontdoor criterion. The combination of these mechanisms ensure that no backdoor paths exist between the input $X$ and the output $Y$ that could bias the training.

**Transportability** We have identified an unbiased interventional query from observational information only. We now study the transportability of each component across domains using the $\mathcal{S}$-admissibility criterion (Jalaldoust & Bareinboim, 2024): conditional probabilities independent from the domain selection variable $S$ are transportable. We assume a training source domain $\mathcal{M}^s$ and a target test domain $\mathcal{M}^t$, it follows graphically that:

$$(W \perp\!\!\!\perp S | \mathcal{D}_X, \mathcal{D}_Y)_{\mathcal{G}^{\Delta_{st}}} \tag{3}$$

$$(R \perp\!\!\!\perp S | X, W, \mathcal{D}_X, \mathcal{D}_Y)_{\mathcal{G}^{\Delta_{st}}} \tag{4}$$

$$(X \not\perp\!\!\!\perp S | \mathcal{D}_X, \mathcal{D}_Y)_{\mathcal{G}^{\Delta_{st}}} \tag{5}$$

$$(Y \not\perp\!\!\!\perp S | X, R, W, \mathcal{D}_X, \mathcal{D}_Y)_{\mathcal{G}^{\Delta_{st}}} \tag{6}$$

The first two quantities are $\mathcal{S}$-admissible and can be written as $P^s(w|\mathcal{D}_{X_s}, \mathcal{D}_{Y_s}) = P^t(w|\mathcal{D}_{X_t}, \mathcal{D}_{Y_t}) = P^*(w|\mathcal{D}_X, \mathcal{D}_Y)$ and $P^s(r|x, w, \mathcal{D}_{X_s}, \mathcal{D}_{Y_s}) = P^t(r|x, w, \mathcal{D}_{X_t}, \mathcal{D}_{Y_t}) = P^*(r|x, w, \mathcal{D}_X, \mathcal{D}_Y)$. The last two quantities are not $\mathcal{S}$-admissible and require to have access to information $\mathcal{D}_{X_t}, \mathcal{D}_{Y_t}$ on the target domain. This is an expected result for $P(x'|\mathcal{D}_X, \mathcal{D}_Y)$. Indeed, the input $X$ directly depends on the distribution of its factors of variations. We must learn a suitable representation of $X$ to use it for inference. Note that this probability is independent of $Y$ and does not require access to labelled data. Under the current SCM problem formulation, $P(y|x', r, w, \mathcal{D}_X, \mathcal{D}_Y)$ is not $\mathcal{S}$-admissible because of the backdoor path between the training data $\mathcal{D}$ and the inference target $Y$. However, in out-of-distribution settings, the training and target domains can differ and the backdoor path may not exist. This is a reasonable expectation as the converse of the $\mathcal{S}$-admissibility criterion does not hold: non $\mathcal{S}$-admissibility does not implies non-transportability. We discuss the limitations of the current formalism in Section 6.

**Tractable Approximation** The quantities above are not directly tractable and must be estimated. The marginalisation over W is a standard technique from Bayesian inference (Magris & Iosifidis, 2023). The integral can be approximated using Markov Chain Monte Carlo (MCMC) or Variational Inference (Blundell et al., 2015b; Magris & Iosifidis, 2023). Specifically:

$$P(y|do(x), \mathcal{D}_X, \mathcal{D}_Y)$$

$$= \int_w P(w|\mathcal{D}_X, \mathcal{D}_Y) \int_r P(r|x, w, \mathcal{D}_X, \mathcal{D}_Y) \int_{x'} P(x'|\mathcal{D}_X, \mathcal{D}_Y) P(y|x', r, w, \mathcal{D}_X, \mathcal{D}_Y) \, dx' \, dr \, dw$$

$$\approx \frac{1}{M} \sum_{j=1}^{M} \int_r P(r|x, w_j, \mathcal{D}_X, \mathcal{D}_Y) \int_{x'} P(x'|\mathcal{D}_X, \mathcal{D}_Y) P(y|x', r, w_j, \mathcal{D}_X, \mathcal{D}_Y) \, dx' \, dr$$

where $w_j \sim P(w|\mathcal{D}_X, \mathcal{D}_Y)$ and M is the amount of weight samples. The distribution of the weights W can be learned by a Bayesian Neural Network (BNN) (MacKay, 1992; Blundell et al., 2015b; Magris & Iosifidis, 2023). The network does not directly learn the weights $w$ but instead estimate the parameters of the distribution in which the weights belong and sample the weights from this distribution at inference. The learning objective cannot be directly optimised because of the high-dimensional and non-convex posterior distribution $P(w|\mathcal{D})$. Instead, MCMC (Izmailov et al., 2021) and Variational Inference (Blundell et al., 2015b) algorithms are preferred. The marginalisation over $R$ is approximated similarly:

$$\int_r P(r|x, w_j, \mathcal{D}_X, \mathcal{D}_Y) \int_{x'} P(x'|\mathcal{D}_X, \mathcal{D}_Y) P(y|x', r, w_j, \mathcal{D}_X, \mathcal{D}_Y) \, dx' \, dr$$

$$\approx \frac{1}{L} \sum_{k=1}^{L} \int_{x'} P(x'|\mathcal{D}_X, \mathcal{D}_Y) P(y|x', r_k, w_j, \mathcal{D}_X, \mathcal{D}_Y) \, dx'$$

where $r_k \sim P(r|x, w_j, \mathcal{D}_X, \mathcal{D}_Y)$ and L is the amount of latent representation samples. Variational Autoencoders (VAEs) (Kingma & Welling, 2014) are commonly used to estimate this distribution. In

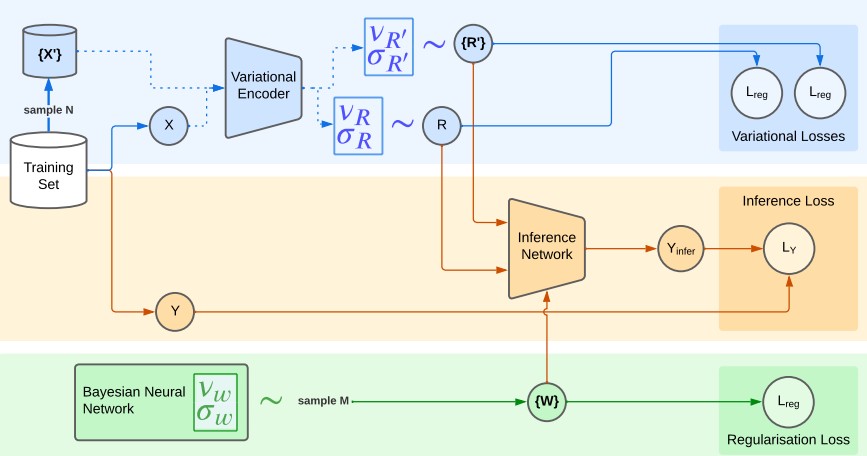

Figure 2: Architecture of the Causal-Invariant Bayesian (CIB) neural network. At the top (in blue), a variational encoder generates the respective parameters of the distributions of the intermediate representations $R$ and $\{R'_i\}_{i=1}^N$ of the input $X$ and the contextual information $\{X'_i\}_{i=1}^N$. $R$ and $\{R'_i\}_{i=1}^N$ are then provided to the inference module (in orange) that retrieves $Y$. This procedure aims to disentangle the learning of the representation $R$ from the learning of the inference mechanisms and force the inference module to only learn the latter. The weights $W$ of the inference module are sampled from a distribution learned using variational inference (in green). The weight sampling and the variational encoding are regularised using an ELBO loss.

practice, we consider a single representation $R$ per input and context image, i.e. $L = 1$. Finally, the last marginalisation can be obtained by sampling input images from the current data distribution:

$$\int_{x'} P(x'|\mathcal{D}_X, \mathcal{D}_Y) P(y|x', r, w_j, \mathcal{D}_X, \mathcal{D}_Y) \, dx' \approx \frac{1}{N} \sum_{k=1}^{N} P(x_i|\mathcal{D}_X, \mathcal{D}_Y) P(y|x_i, r_k, w_j, \mathcal{D}_X, \mathcal{D}_Y)$$

where N is the amount of input image samples.

## 4 CAUSAL-INVARIANT BAYESIAN NEURAL NETWORK

We now propose our Causal-Invariant Bayesian (CIB) architecture estimating the posterior probability defined above. The architecture is summarised in Figure 2. It can be decomposed into three components:

- A variational encoder (in blue) learning domain-invariant information $R$ by jointly optimising the inference loss with an ELBO loss,
- An inference module (in orange) combining domain-specific and domain-invariant information to solve the task,
- A weight distribution $W$ (in green) learned by optimising the inference objective with an ELBO loss and from which are sampled the parameters of the inference module.

**Variational Encoder** The variational encoder is used to learn the intermediate domain-invariant representations $R$ and $\{R'_i\}_{i=1}^N$. The network weights are learned by optimising an estimate lower bound (ELBO) loss. Existing work attempting to disentangle domain-specific and domain-invariant features typically use a Variational Autoencoder (VAE) and add a reconstruction loss (Gabbay et al., 2021; Yang et al., 2021; Zhang et al., 2022). The input and the contextual information are fed to two different encoders, learning to specialise to invariant features and domain-specific features, respectively. However, we find that a single encoder can yield better performance. By jointly giving

the input and the context to the encoder, we teach it to discard irrelevant domain-specific features at once. We use a pre-trained ResNet-18 (He et al., 2016) without the final classification layer as the encoder. For comparison, we perform additional experiments with the Causal Transportability architecture that integrates a VAE and a reconstruction loss (Mao et al., 2022). Following the methodology of partially-stochastic neural networks (Sharma et al., 2023), we use a point-estimate network for the encoder and only use a Bayesian network for the inference network. This choice allows to maintain the expressivity on the level of uncertainty inherent to Bayesian neural networks while having a limited impact on the optimisation efficiency.

**Bayesian Inference Network**  For the inference model, we use a dense network with stochastic layers for the input and output and a single point-estimate hidden layer. The inference module takes batches of $N$ inputs. Each input is a sum of the input representation $R$ and one context representation $R_i'$. We select the mean of the results with respect to $N$ to estimate the marginalisation over the context. We use a reparametrisation trick (Blundell et al., 2015a) to maintain the full differentiability of the network. Following Elfwing et al. (2018), we change the activation layers from ReLU to SiLU.

**Context Regularisation**  We add label-mixing regularisation to complement the contextual information during training, following the procedure used in Mixup (Zhang et al., 2018) and Manifold Mixup (Verma et al., 2019). We mix the true label $\mathbf{y_t}$ with the labels $\mathbf{Y_{x_i}}$ of the $N$ context images $x_i$:

$$\mathbf{y} = \alpha \cdot \mathbf{y_t} + (1 - \alpha) \cdot \frac{1}{N} \sum_{i=1}^{N} \mathbf{y_{x_i}} \tag{7}$$

**Weight Function Regularisation**  We further regularise the weights sampling of the inference module to enhance the diversity of the weights while aligning it with functional diversity. Following the work of Roy et al. (2024), we add a regularisation loss inducing the network to assign the same posterior distribution to weights realising an identical function. For a batch of inputs $\mathbf{x}$ and a set of weight samples $\mathcal{W} = \{w_i\}_{i=1}^{M}$, we define our *weight function regularisation* loss in Equation 8. $\text{Comb}(\mathcal{W}, 2)$ denotes the set of 2-combinations from $\mathcal{W}$.

$$\mathcal{L}_{weight\_func} = \sum_{(w_i, w_j) \in \text{Comb}(\mathcal{W}, 2)} ||f_{w_i}(\mathbf{x}) - f_{w_j}(\mathbf{x})||^2 \tag{8}$$

**Loss function**  The final loss function comprises the mixed labels loss, the weight function regularisation and the regularisation terms of the ELBO losses. The mixed label loss is a cross entropy loss between the predicted distribution and the true mixed distribution as described above. We add the weight function regularisation loss described in the previous paragraph, weighted by a hyperparameter $\beta$. Then, KL divergence losses are added for regularising the variational parameters for the input representation ($\nu_R, \sigma_R$), the context representation ($\nu_{R'}, \sigma_{R'}$) and the Bayesian weights ($\nu_W, \sigma_W$). These additional losses are weighted by their respective hyperparameters $\gamma$, $\mu$ and $\epsilon$. The complete loss is shown in Equation 9.

$$\begin{aligned} \mathcal{L} = \text{CrossEntropy}(\mathbf{y_{pred}}, \mathbf{y}) + \beta \cdot \mathcal{L}_{weight\_func} + \gamma \cdot \text{KL}(\mathcal{N}(\nu_R, \sigma_R)||\mathcal{N}(0, 1)) \\ + \mu \cdot \text{KL}(\mathcal{N}(\nu_{R'}, \sigma_{R'})||\mathcal{N}(0, 1)) + \epsilon \cdot \text{KL}(\mathcal{N}(\nu_W, \sigma_W)||\mathcal{N}(0, 1)) \end{aligned} \tag{9}$$

## 5 EXPERIMENTS

### 5.1 DATASETS

We perform experiments on image recognition tasks. All datasets contain train, validation and test splits in-distribution (i.i.d) and out-of-distribution (o.o.d). We first conduct image recognition on the image recognition CIFAR10 dataset (Krizhevsky et al., 2009). We build the o.o.d sets by performing random translations of the input images. We also use the OFFICEHOME dataset (Venkateswara et al., 2017) which contains four subsets of images containing objects in various configurations (real world, product sheet, art, clipart). We train on one configuration and evaluate o.o.d in another configuration.

Table 1: Accuracy on CIFAR-10. The CIFAR-10 column shows the results on the i.i.d test set while the columns on the right show the results with an increasing level of perturbation of the test set. The mean and standard deviation across three runs are shown. Our model outperforms the baselines and the ablated models. Particularly, mixup information has a significant impact on the final accuracy.

| | CIFAR-10 | CIFAR-10-o.o.d-perturb. | | |
| --- | --- | --- | --- | --- |
| | | 0.1 | 0.2 | 0.4 |
| ResNet-18 | $0.739 \pm 0.005$ | $0.599 \pm 0.017$ | $0.471 \pm 0.022$ | $0.284 \pm 0.010$ |
| ResNet-18-CT | $0.682 \pm 0.004$ | $0.507 \pm 0.010$ | $0.397 \pm 0.006$ | $0.244 \pm 0.010$ |
| -w/-pretrainedVAE | $0.641 \pm 0.004$ | $0.464 \pm 0.017$ | $0.358 \pm 0.015$ | $0.229 \pm 0.010$ |
| CIBResNet-18 (ours) | $\mathbf{0.763 \pm 0.004}$ | $\mathbf{0.646 \pm 0.006}$ | $\mathbf{0.510 \pm 0.012}$ | $0.301 \pm 0.006$ |
| -w/o-mixup | $0.761 \pm 0.004$ | $0.625 \pm 0.013$ | $0.494 \pm 0.009$ | $0.297 \pm 0.004$ |
| -w/o-weight-func | $0.762 \pm 0.009$ | $0.635 \pm 0.010$ | $0.505 \pm 0.009$ | $\mathbf{0.305 \pm 0.007}$ |

## 5.2 Experimental Setup

We compare our CIBResNet-18 against ResNet-18 and ResNet-18-CT in i.i.d and o.o.d settings. The second baseline is the causal-Transportability (CT) algorithm of Mao et al. (2022). Following their methodology, we use a VAE for the $P(R|X)$ encoder. Their proposed inference network is a three-layers convolution network followed by a two-layers dense classifier. For a fair comparison, we use a modified ResNet-18 (the same size as our model) that takes input and contextual information. More details are given in Appendix B. We conduct our experiments on a single GPU Nvidia A100 with the AdamW optimiser (Loshchilov & Hutter, 2019). We initialise the Bayesian classifier parameters from the Normal distribution $\mathcal{N}(0, 1)$. On CIFAR-10, we train our model and the baselines for 20 epochs. We find that all models usually do not show improvements after 10 epochs. On OFFICEHOME, we train the models for 80 epochs. The training sets contain less samples than CIFAR-10 and more epochs are required for convergence. ResNet-18 stops improving after 40 epochs while CIBResNet-18 keep improving until around 60 epochs. We use hyperparameter grid search on the validation set of CIFAR-10 to set the hyperparameters. We use the default learning rate of 0.005 for the baselines and 0.01 for CIBResNet-18. We use a batch size of 64. By default, we use $N = 16$, $M = 16$, $\alpha = 0.4$, $\beta = 0.01$, $\gamma = 10^{-6}$, $\nu = 10^{-6}$ and $\epsilon = 10^{-6}$. The remaining hyperparameters are set to their default values. When evaluating the models in o.o.d settings, we use the batch statistics to build the mean and variance in the batch normalisation layers instead of the values learned on the training set that follow a different distribution.

## 5.3 Visual Recognition

We compare our model against ResNet-18 and ResNet-18-CT on CIFAR10. The results are shown in Table 1. Curiously, the base ResNet-18 outperforms the ResNet-18-CT models. This result reinforces our hypothesis that a reconstruction loss is not necessary and can even be adversarial in some situations. We further study how our model behaves on a more challenging dataset and compare it with ResNet-18. Figure 3 shows the results on OFFICEHOME. Our model systematically outperforms the baselines. Ablation studies in Table 1 demonstrate the importance of adding contextual mixup information to the label during training to help the model incorporate the context images during learning. We further compare the training of our model with the baseline on CIFAR-10 and the first domain of OFFICEHOME in Figure 4. We observe that the validation accuracy of the baselines first decreases to reach a plateau but then increases again, highlighting an overfitting to the training data. However, this behaviour is not observed with CIBResNet-18, which steadily decreases across the entire training, demonstrating higher stability. We can also note that our proposed model yields a lower standard deviation in the accuracy.

## 5.4 Impact of Sampling on Posterior

We perform additional experiments to investigate the impact of the number of samples. We vary the amounts $N$ and $M$ of context images and inference weights, respectively. Figure 5 shows the evolution of the accuracy of our model when varying $N$ and $M$. The main contributing factor for performance is the number of context samples: using four context samples instead of one significantly

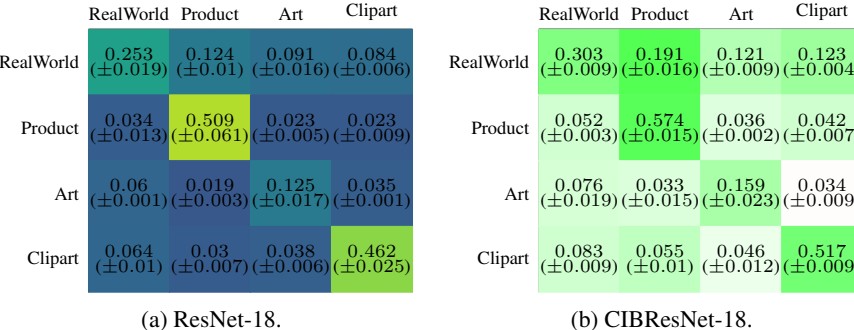

(a) ResNet-18.  (b) CIBResNet-18.

Figure 3: Domain transfer results on the OFFICEHOME dataset. Each row represents the category of the training subset and each column represents the category of the test subset. Accuracy with a random guess is 0.015. In the right figure, a cell is shown in green if its value is higher than the baseline on the left. The mean and standard deviation across three runs are shown. Our proposed model (on the right) systematically outperforms the baseline (on the left). Only the model trained on the Art domain shows little to no improvement. As this domain demonstrates the lowest i.i.d and o.o.d accuracy, we explain it by the lack of exploitable training data.

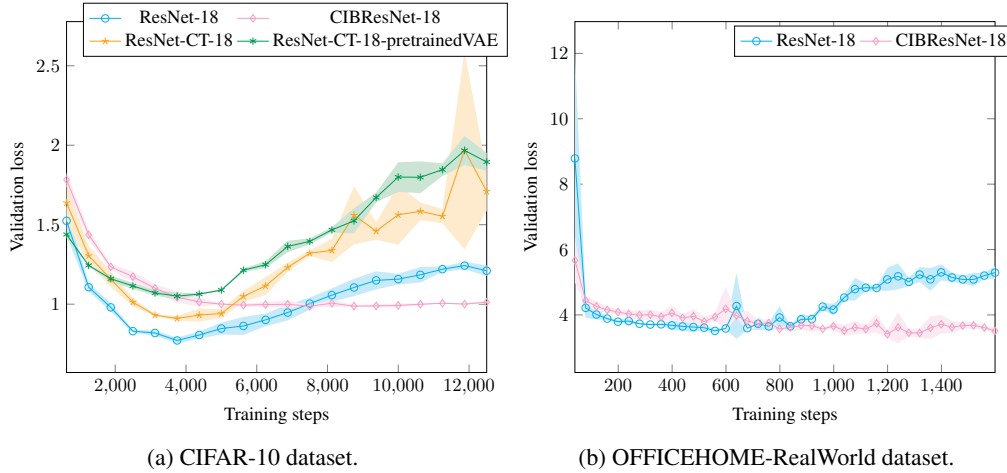

(a) CIFAR-10 dataset.  (b) OFFICEHOME-RealWorld dataset.

Figure 4: Evolution of the validation loss during training. The mean and standard deviation across three runs are shown. After a period of improvement, the ResNet-18 and ResNet-18-CT baselines overfit as the training progresses. This behaviour is not observed with CIBResNet-18, which also demonstrates a lower standard deviation.

improves accuracy. We hypothesize that a lower number is not sufficient to be representative of the distribution and has an adversarial effect instead. Figure 6 shows the evolution of the accuracy during training for several values of $N$ and $M$. We observe that increasing the number of weight samples $M$ improves sample efficiency and reduces the amount of training steps required to converge.

## 6 LIMITATIONS

This study focuses on developing a theory of interventional queries for o.o.d tasks and implementing an architecture suitable for answering such queries. We do not focus on the optimisation and the efficiency of such a system. Bayesian neural networks are notoriously difficult to optimise (Magris & Iosifidis, 2023). As discussed in the experimental section, the sampling step can hurt the efficiency of the model. As argued by Hooker (2021), the available hardware has a great impact on the success of a machine learning method. The current hardware landscape is well suited for point estimate neural networks but not for Bayesian and variational methods requiring many sampling steps. We argue that developing better and usable Bayesian causal models will require, not only improving the theory, but

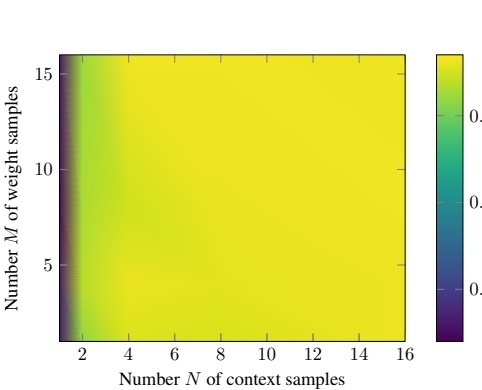

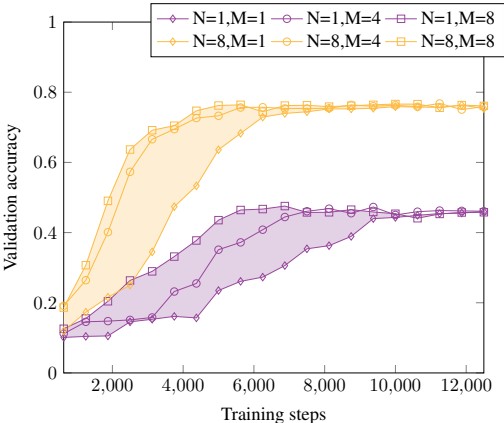

Figure 5: Accuracy heatmap of CIBResNet-18 on the CIFAR-10 test set as a function of the number of weight and context samples. The amount of weight samples has a negligible effect on performance. Only increasing the context size improves accuracy.

Figure 6: Evolution of the validation accuracy of CIBResNet-18 on CIFAR-10 during training with varying context images $N$ and weight samples $M$. Increasing the weight samples reduces the amount of training steps required for learning.

more importantly developing new optimisation strategies that can bridge the efficiency gap with their point estimate counterparts.

We perform our analysis under the SCM framework (Pearl, 2009). The underlying causal model is represented by a DAG that does not allow feedback loops. As the training of a neural network is an iterative process, a representation that allows cycles could better represent the inner causal mechanisms behind the learning process and help improve generalisation. This is a challenging task (Bongers et al., 2021) that we will tackle in our future work.

## 7 CONCLUSION

We propose a Causal-Invariant Bayesian neural network architecture to improve domain generalisation on visual recognition tasks. We build a theoretical analysis for domain generalisation with deep learning based on causality theory and taking into account the parameter learning process. We show experimentally that following causal principles can improve robustness to distribution shifts and reduce overfitting. These problems are particularly prominent on strong reasoning tasks that require neural systems to use abstract domain-invariant mechanisms and ignore spurious domain-specific information (Chollet, 2019; Zhang et al., 2021). In our future work, we will extend our framework to solve these challenging reasoning tasks. This investigation also provides elements to motivate the development of a causality-based theory of deep learning following the Independent Causal Mechanisms principle (Peters et al., 2017; Schölkopf et al., 2021). Such theory could improve robustness and generalisation in neural networks and provide additional benefits such as improved interpretability and modularity.

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

## A    Proofs for the Interventional Bayesian Inference

### A.1    Rules of do-calculus

We first describe the three rules of do-calculus (Pearl, 2009) as they are required for the proofs in Sections A.2 and A.3. The first rule states that an observation $z$ can be ignored if it does not affect the outcome $y$ of the query given the current causal graph. The second rule states that an intervention on a variable $do(z)$ can be considered an observation $z$ if there are no backdoor paths linking it to the outcome $y$, i.e. if the variables $Z$ and $Y$ do not share any common ancestors not captured by the observation $z$. The third rule states that an intervention $do(z)$ can be ignored if there are no direct paths between $Z$ and $Y$ or backdoor paths between $Y$ and observed descendants $W$ of $Z$.

The three rules are formally defined as follows. For each rule, the equality holds if the independence test between the variable of interest $Z$ and the outcome $Y$ is verified in the given causal graph. The graph $\mathcal{G}_{\overline{X}}$ corresponds to the initial graph with the incoming links to $X$ removed, corresponding to an intervention on $X$. The graph $\mathcal{G}_{\overline{X}}$ furthermore removes the outgoing links of $Z$. The term $\overline{Z(W)}$ removes the incmoing links of $Z$ if $Z$ is not an ancestor of the observed variable $W$.

- **Rule 1** Deletion of observation

$$P(y|do(x), z, w) = P(y|do(x), w) \text{ if } (Y \perp\!\!\!\perp Z|X, W)_{\mathcal{G}_{\overline{X}}} \tag{10}$$

- **Rule 2** Reduction of intervention to observation

$$P(y|do(x), do(z), w) = P(y|do(x), z, w) \text{ if } (Y \perp\!\!\!\perp Z|X, W)_{\mathcal{G}_{\overline{X}\underline{Z}}} \tag{11}$$

- **Rule 3** Deletion of intervention

$$P(y|do(x), do(z), w) = P(y|do(x), w) \text{ if } (Y \perp\!\!\!\perp Z|X, W)_{\mathcal{G}_{\overline{X}, \overline{Z(W)}}} \tag{12}$$

### A.2    Identifiability of $P(y|do(x), \mathcal{D}_X, \mathcal{D}_Y)$

*Proof.* Proof of Equation 2. We start by marginalising on W to block the backdoor path between R and Y ($R \leftarrow U_R \rightarrow \mathcal{D}_R \rightarrow W \rightarrow Y$) and use the rules of do-calculus to simplify the quantities. This step ensures that we can later apply the frontdoor criterion with R and block the backdoor paths between X and Y.

$$P(y|do(x), \mathcal{D}_X, \mathcal{D}_Y)$$
$$= \int_w P(y|do(x), \mathcal{D}_X, \mathcal{D}_Y, w) P(w|do(x), \mathcal{D}_X, \mathcal{D}_Y) \, dw \qquad \textit{Marginalisation over W}$$
$$= \int_w \boxed{P(y|do(x), \mathcal{D}_X, \mathcal{D}_Y, w)} P(w|\mathcal{D}_X, \mathcal{D}_Y) \, dw \qquad \textit{Rule 3: } (W \perp\!\!\!\perp X|\mathcal{D}_X, \mathcal{D}_Y)_{\mathcal{G}_{\overline{X}}}$$

We now focus on the quantity in the red box. We must use the frontdoor criterion to block all backdoor paths between X and Y: $X \leftarrow U_{XY} \rightarrow Y$, $X \leftarrow Z \rightarrow \mathcal{D}_X \leftarrow U_{XY} \rightarrow Y$. We marginalise on R.

$$\boxed{P(y|do(x), \mathcal{D}_X, \mathcal{D}_Y, w)}$$
$$= \int_r P(y|do(x), \mathcal{D}_X, \mathcal{D}_Y, w, r) P(r|do(x), \mathcal{D}_X, \mathcal{D}_Y, w) \, dr \qquad \textit{Marginalisation over R}$$
$$= \int_r \boxed{P(y|do(x), \mathcal{D}_X, \mathcal{D}_Y, w, r)} P(r|x, \mathcal{D}_X \mathcal{D}_Y, w) \, dr \qquad \textit{Rule 2: } (R \perp\!\!\!\perp X|\mathcal{D}_X, \mathcal{D}_Y, W)_{\mathcal{G}_{\underline{X}}}$$

Again, we focus on the quantity in the blue box. We continue to apply the frontdoor criterion and marginalise on X after switching the intervened variable using do-calculus rules.

$$\boxed{P(y|do(x), \mathcal{D}_X, \mathcal{D}_Y, w, r)}$$

$$= P(y|do(x), do(r), \mathcal{D}_X, \mathcal{D}_Y, w) \qquad\qquad \textit{Rule 2: } (Y \perp\!\!\!\perp R|X, \mathcal{D}_X, \mathcal{D}_Y, W)_{\mathcal{G}_{\overline{X}\underline{R}}}$$

$$= P(y|do(r), \mathcal{D}_X, \mathcal{D}_Y, w) \qquad\qquad\qquad \textit{Rule 3: } (Y \perp\!\!\!\perp X|R, \mathcal{D}_X, \mathcal{D}_Y, W)_{\mathcal{G}_{\overline{XR}}}$$

$$= \int_{x'} P(y|do(r), \mathcal{D}_X, \mathcal{D}_Y, w, x')P(x'|do(r), \mathcal{D}_X, \mathcal{D}_Y, w)\, dx' \qquad\qquad \textit{Marginalisation over X}$$

$$= \int_{x'} P(y|r, \mathcal{D}_X, \mathcal{D}_Y, w, x')P(x'|do(r), \mathcal{D}_X, \mathcal{D}_Y, w)\, dx' \qquad \textit{Rule 2: } (Y \perp\!\!\!\perp R|X, \mathcal{D}_X, \mathcal{D}_Y, W)_{\mathcal{G}_{\underline{R}}}$$

$$= \int_{x'} P(y|r, \mathcal{D}_X, \mathcal{D}_Y, w, x')P(x'|\mathcal{D}_X, \mathcal{D}_Y, w)\, dx' \qquad\qquad \textit{Rule 3: } (X \perp\!\!\!\perp R|\mathcal{D}_X, \mathcal{D}_Y, W)_{\mathcal{G}_{\overline{R}}}$$

$$= \int_{x'} P(y|r, \mathcal{D}_X, \mathcal{D}_Y, w, x')P(x'|\mathcal{D}_X, \mathcal{D}_Y)\, dx' \qquad\qquad\qquad \textit{Rule 1: } (X \perp\!\!\!\perp W|\mathcal{D}_X, \mathcal{D}_Y)_{\mathcal{G}}$$

We put together all the quantities and obtain Equation 2:

$$P(y|do(x), \mathcal{D}_X, \mathcal{D}_Y)$$
$$= \int_w P(w|\mathcal{D}_X, \mathcal{D}_Y) \int_r P(r|x, w, \mathcal{D}_X, \mathcal{D}_Y) \int_{x'} P(x'|\mathcal{D}_X, \mathcal{D}_Y)P(y|x', r, w, \mathcal{D}_X, \mathcal{D}_Y)\, dx'\, dr\, dw$$

$$\qquad\qquad\qquad\qquad\qquad\qquad\qquad\qquad\qquad\qquad\qquad\qquad\qquad\qquad\qquad\qquad\square$$

### A.3 Intractability of $P(y|do(x), do(\mathcal{D}_X), \mathcal{D}_Y)$

*Proof.* Proof of the intractability of $P(y|do(x), do(\mathcal{D}_x), \mathcal{D}_y)$. We show that estimating this query from observational data requires to marginalise on $\mathcal{D}_X$. We take as a postulate that this is an intractable operation because it implies to access all possible input domains. We also note that estimating this operation via sampling can be achieved but goes against the purpose of domain generalisation as we aim to adapt to new domains from limited information.

From the causal graph in Figure 1b, we observe two causal paths linking $\mathcal{D}_X$ to $Y$ :

1. A direct path: $\mathcal{D}_X \to \mathcal{D}_R \to W \to Y$

2. A backdoor path: $\mathcal{D}_X \leftarrow Z \to X \to R \to Y$

3. A backdoor path: $\mathcal{D}_X \leftarrow U_{XY} \to Y$

The third path is blocked by the do operation on $X$ but the first two paths realise the frontdoor criterion as shown in the simplified causal graph in Figure 7. The expression can therefore be simplified as follows:

$$P(y|do(x), do(\mathcal{D}_X), \mathcal{D}_Y)$$
$$= \int_a P(a|\mathcal{D}_X, do(x), \mathcal{D}_Y) \int_{\mathcal{D}'_X} P(y|do(x), \mathcal{D}'_X, \mathcal{D}_Y)P(\mathcal{D}'_X|do(x), \mathcal{D}_Y)\, d\mathcal{D}'_X \quad \textit{frontdoor criterion}$$

Answering this query from observations requires marginalising on $D_X$, as highlighted in blue.

We can see graphically that this path cannot be blocked by any other variable than $\mathcal{D}_X$ because the variable $U_{XY}$ is not observed, therefore marginalising on $\mathcal{D}_X$ is necessary.

$$\qquad\qquad\qquad\qquad\qquad\qquad\qquad\qquad\qquad\qquad\qquad\qquad\qquad\qquad\qquad\qquad\square$$

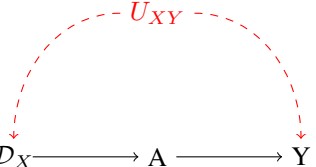

Figure 7: Simplified causal graph linking the input domain $\mathcal{D}_X$ to the output label $Y$. The paths can be simplified as a backdoor path via $U_{XY}$ and a direct path through an an abstract variable $A$.

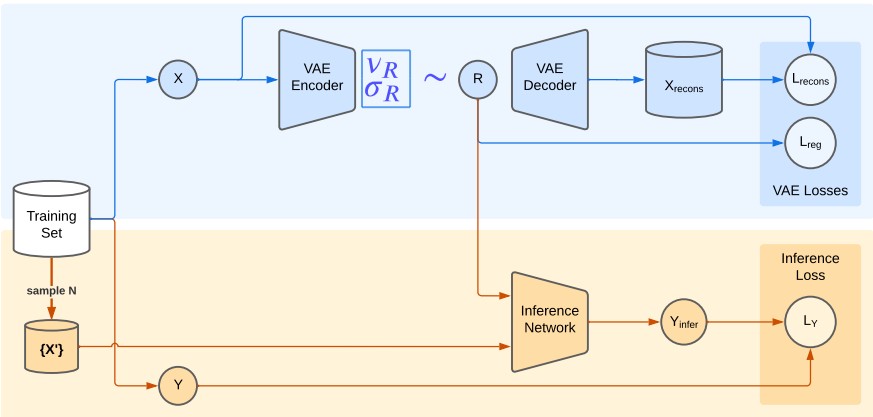

Figure 8: Illustration of the Causal Transportability baseline model.

## B    DETAILS ON THE CAUSAL TRANSPORTABILITY MODEL

We implement the Causal Transportability algorithm defined by Mao et al. (2022) as a baseline. The model is illustrated in Figure 8. The algorithm realises the following quantity:

$$P(y|do(x)) = P(r|x) \sum_{k=1}^{N} P(y|r, x_i) P(x_i) \tag{13}$$

The sampling strategy is the same as the one used in our work. The authors use a small network composed of three convolution layers followed by two fully connected layers to realise the quantity $P(y|r, x_i)$. For a fair comparison with our model, we replace it with a modified ResNet-18. We alter the first layer is modified to take a concatenation of an input image $x$ with the representation $r$. The rest of the network is left untouched. The quantity $P(r|x)$ can be obtained via several methods. We use a VAE in our implementation, as the main method described in the original paper. We use the model in two settings. First, we jointly train the VAE and the inference network. Second, we compare this baseline against a second model where we train the VAE separately on a reconstruction task and then include the weights to the full model. We allow the training of the VAE weights at the second stage as we observed that freezing the weights leads to decreased performance.

