# OpenReview forum: "Robust Domain Generalisation with Causal Invariant Bayesian Neural Networks"
_ICLR.cc/2025/Conference — Submitted to ICLR 2025_

### Official Review · Reviewer_uvMu · 2024-10-27

**Soundness:** 2
**Presentation:** 3
**Contribution:** 2
**Rating:** 3
**Confidence:** 3

**Summary:**

This paper exploits the invariance of causal mechanisms under distribution changes to improve out-of-distribution generalisation.
It proposes the Causal-Invariant Bayesian (CIB) neural network. The CIB architecture combines a variational encoder, an inference module, and a Bayesian neural network,
aiming to learn domain-invariant representations. Theoretically, the model is shown to approximate reasoning under causal interventions.
Experimentally, the model improves out-of-distribution generalisation in image recognition tasks.

**Strengths:**

1. It is interesting to exploit the causal mechanism invariance to improve out-of-distribution generalisation.
2. Both theoretical and empirical evidence are provided to validate the proposed architecture.
3. The formulations are clearly presented and illustrated.

**Weaknesses:**

1. It is not a new idea to exploit the causal mechanism invariance to improve out-of-distribution generalisation.
Many works have already used this idea to improve out-of-distribution generalization, e.g., [1] [2] [3] [4].
The paper does not discuss these related works sufficiently and compare the proposed method to these approaches.
2. The experiments only compare a single method for out-of-distribution generalization, which does not even appear to outperform the vanilla baseline (Table 1).
The comparison is insufficient to prove the superiority of the proposed method.
3. The proposed architecture seems to be an integration of several existing methods.



[1] De Haan, Pim, Dinesh Jayaraman, and Sergey Levine. "Causal confusion in imitation learning." NeurIPS, 2029.

[2] Zhang, Amy, Clare Lyle, Shagun Sodhani, Angelos Filos, Marta Kwiatkowska, Joelle Pineau, Yarin Gal, and Doina Precup. "Invariant causal prediction for block mdps." ICML, 2020.

[3] Bica, Ioana, Daniel Jarrett, and Mihaela van der Schaar. "Invariant causal imitation learning for generalizable policies." NeurIPS, 2021.

[4] Liu, Chang, Xinwei Sun, Jindong Wang, Haoyue Tang, Tao Li, Tao Qin, Wei Chen, and Tie-Yan Liu. "Learning causal semantic representation for out-of-distribution prediction." NeurIPS, 2021.

**Questions:**

1. What are the novelty and the superiority of the proposed method over other methods that enhance out-of-distribution generalization through causal mechanism invariance?
2. What are the novel aspects of the proposed architecture beyond integrating existing techniques?

---

> ### Author Response · Authors · 2024-11-23
>
> Thank you for your review and for pointing out the lack of references on existing domain generalisation studies! We acknowledge that our works lacks comparisons with other methods. We have integrated this feedback in our paper and added comparisons between our method and the existing, highlighting the conceptual differences between the methods and conducted comparative experiments on standard benchmarks.
>
>
> Regarding your questions:
>
> 1. We have added comparison with the missing references. Please, refer to the general comment above to see the details. We are currently running the experiments and hope to add them before the end of the discussion phase.
>
> 2. Our work builds upon causal representation learning techniques for inducing domain-invariant mechanisms in neural networks. As such, our work aims to train the model to compute interventional queries P(y|do(x)). It is motivated by the observation that existing studies do not consider spurious correlations arising by the conditioning on the training set (as shown in Figure 1). From this observation, we derive a quantity that allows learning interventional queries while taking this bias into account. This derivation is the main contribution of our work and our proposed architecture is a direct result from it. Additional differences with existing work are highlighted in the general comment at the top.
>
> Please, let us know if our response answers your concerns and if you would consider raising your score in the light of additional experiments and baseline comparisons. Otherwise, let us know what else we should improve.

---

> > ### Comment · Reviewer_uvMu · 2024-11-26
> >
> > Thank you for your response.
> >
> > I remain unconvinced about the novelty of the work. On the one hand, the general comment states "[7,8,9,10,11] are studies applying causality theory and Bayesian methods to different problems than the one considered in our work"; on the other hand, it looks like the work aims to propose a general learning method rather than focus on a certain problem. In this light, the current discussion of related work does not sufficiently establish how the proposed method is novel compared to those cited.
> >
> > Additionally, the work claims to rely on an interventional quantity rather than domain-invariant information. However, isn't domain invariance the best we can hope for in achieving robust domain generalization?

---

> > > ### Author Response · Authors · 2024-11-26
> > >
> > > Thank you for your response. We understand that our initial comment did not provide enough information on the differences between our work and [7,8,9,10,11].
> > >
> > > Regarding the scope of our work: although the theoretical foundations of our work are not specific to image classification, the implementation we propose is tailored for this task. Extending it to other tasks is a challenging open problem due to the specificities of the possible modalities involved. For example, the causal variables $X$ and $Y$ are not elucidated in our theorical section but their modalities have a high impact on the downstream architecture: the marginalisation over the domain of $X$ can be estimated by sampling because instances of $X$ are images. Other modalities may have better strategies, e.g. inputs of smaller dimensions may not require sampling but could be directly estimated. Similarly, the probability $P(Y|do(X))$ can be directly estimated because $Y$ is a discrete class variable. Other modalities (e.g. continuous values or image segmentations) would require additional research to create different adaptations. Therefore, we follow the approach by previous work on o.o.d generalisation and focus on domain shifts with image classification.
> > >
> > > Regarding the specific differences between our work and existing approaches: [7] augments input images using randomly initialised convolutional networks and trains a segmentation model to make the same prediction on the augmented version of the input image. While our approach differs as we use in-domain sampling, this augmentation approach is similar to that of [2], although this augmentation technique only modifies image intensity and texture and does not consider changes in shape or translations/rotations. This is well suited for medical images and for segmentation tasks as the augmentation does not alter the ground-truth segmentation mask, but not for the more general case considered in our work.
> > >
> > > [8] proposes a prompt learning method for image-language models. The aim of the paper is not to train a parametric model to solve a task but to optimise the input prompt given to a language model to boost its performance. To regularise training and reduce overfitting, the authors formulate the problem as estimating the prompt space distribution and use a Bayesian method to solve it. The problem settings are very different as the target distribution is a general prior over the possible prompts while we aim to compute an (interventional) posterior probability $P(Y|do(X))$.
> > >
> > > [9] tackles the problem of causal misidentification in imitation learning. The proposed method attempts to learn the true causal model behind the policy of experts acting in an environment. This problem differs from ours as it aims to discover the complete causal structure underlying a policy while we aim to elucidate one causal query given a known causal structure (see Figure 1). More importantly, this work assumes access to interventional data from the environment, either by querying an expert or by directly acting in the environment, while we are in a supervised settings and only have access to a fixed set of observations. [10] and [11] are similar reinforcement learning methods and share the same differences.
> > >
> > > Thank you for pointing out our lack of clarity, we hope that this second comment elucidates the differences with our work. We added this information to our general comment above.

---

> > > > ### Author Response · Authors · 2024-11-26
> > > >
> > > > Regarding your second question: this is a good point! A majority of the previous work assumes that only domain-invariant information should be learned by the network (e.g. [1] learns $P(Y|Z_R)$). However, recent work has pointed out that domain-specific information remains helpful for the task and that the issue lies in the entanglement of the two mechanisms (e.g. [2]). As an example, if we train a classifier on two domains $\mathcal{D}_1$ and $\mathcal{D}_2$ that have different and very imbalanced label distributions (e.g. $\mathcal{D}_1$ largely favours label 1 and $\mathcal{D}_2$ largely favours label 2), keeping domain information is useful for the classification. Interventional queries effectively disentangle prior domain knowledge from inference mechanisms, taking advantage of all the available information [12]. We also show using transportability theory [13] (Section 3, transportability paragraph) that, from the causal structure induced by supervised learning, some components of the inference network cannot be made domain-invariant so we have to use domain knowledge in the computation.
> > > >
> > > > We hope this answers your question on the motivations behind our problem formulation. We will make it clearer in the paper. This is an interesting point and we are happy to discuss this topic further.
> > > >
> > > >
> > > > [1] Liu, Chang, Xinwei Sun, Jindong Wang, Haoyue Tang, Tao Li, Tao Qin, Wei Chen, and Tie-Yan Liu. "Learning causal semantic representation for out-of-distribution prediction." NeurIPS, 2021.
> > > >
> > > > [2] Lv, Fangrui, et al. "Causality inspired representation learning for domain generalization." Proceedings of the IEEE/CVF conference on computer vision and pattern recognition. 2022.
> > > >
> > > > [7] Ouyang, Cheng, et al. "Causality-inspired single-source domain generalization for medical image segmentation." IEEE Transactions on Medical Imaging 42.4 (2022): 1095-1106.
> > > >
> > > > [8] Derakhshani, Mohammad Mahdi, et al. "Bayesian prompt learning for image-language model generalization." Proceedings of the IEEE/CVF International Conference on Computer Vision. 2023.
> > > >
> > > > [9] De Haan, Pim, Dinesh Jayaraman, and Sergey Levine. "Causal confusion in imitation learning." NeurIPS, 2029.
> > > >
> > > > [10] Zhang, Amy, Clare Lyle, Shagun Sodhani, Angelos Filos, Marta Kwiatkowska, Joelle Pineau, Yarin Gal, and Doina Precup. "Invariant causal prediction for block mdps." ICML, 2020.
> > > >
> > > > [11] Bica, Ioana, Daniel Jarrett, and Mihaela van der Schaar. "Invariant causal imitation learning for generalizable policies." NeurIPS, 2021.
> > > >
> > > > [12] Pearl, J. (2009). Causality. Cambridge university press.
> > > >
> > > > [13] Jalaldoust, K., & Bareinboim, E. (2024, March). Transportable representations for domain generalization. In Proceedings of the AAAI Conference on Artificial Intelligence (Vol. 38, No. 11, pp. 12790-12800).

---

### Official Review · Reviewer_GRc5 · 2024-11-02

**Soundness:** 3
**Presentation:** 3
**Contribution:** 3
**Rating:** 6
**Confidence:** 3

**Summary:**

This paper proposes an intervention mechanism that can be added to a bayesian inference pipeline to solve out of distribution tasks. The method improves in distribution and out of distribution performance in an unsupervised manner. Intervention is made by leveraging contextual information from within dataset using Mixup strategies. The inference network is a partially stochastic bayesian neural network. Theoretical result is derived for when intervention is made taking into account conditioning on the training dataset. This result is then used to show performance improvement on CIFAR10 and OFFICEHOME datasets.

**Strengths:**

1. The novelty of the paper is to apply causal interventions in a bayesian inference pipeline and show improvements on out of distribution data compared to point estimate based networks.
2. Another novel aspect compared to baseline used is considering datasets in the causal graph and deriving the theoretical result and its execution with partially stochastic networks.
3. The paper provides a methodology to learn causal representations in unsupervised manner.
4. The underlying concepts for bayesian networks, causality are adequately explained.
5. The authors provide an anonymized code repository for review.

**Weaknesses:**

1. Experiments -  The authors show improvements on dataset with translation of CIFAR10 and OFFICEHOME dataset. Results on datasets with other types of commonly seen o.o.d. variations like different backgrounds would make a more convincing point.
2. Visualizations for results would be helpful to understand the domain gap and analyze improvements with proposed method.
3. While the architecture is well described in the paper, the training and evaluation algorithm could be described with more clarity and details for how the theoretical result is used.

**Questions:**

1. How much is the performance improvement due to proposed intervention design vs from the use of bayesian inference over point-estimate methods?
2. While other files are present, the trainer.py file in anonymized code repo is empty. Would appreciate access to understand the method better.

---

> ### Author Response · Authors · 2024-11-23
>
> Thank you for the positive and very detailled review! We are happy that you find our paper of interest. The lack of experiments and comparisons has been pointed out by the other reviewers so we are currently running additional experiments. We hope to add them before the end of the discussion phase.
>
>
> Regarding your questions:
>
> 1. We conduct ablation studies in Figures 5 and 6 that show that the use of BNNs has little impact on the final performance but helps improving sample efficiency during training.
>
> 2. Thank you for pointing this out! Fortunately, the trainer.py file is an empty legacy file unused in the latest version of the code. It will be removed in the future. The full code of the method is available in the provided repository.

---

### Official Review · Reviewer_Ee74 · 2024-11-02

**Soundness:** 1
**Presentation:** 1
**Contribution:** 2
**Rating:** 3
**Confidence:** 5

**Summary:**

This paper proposes a Bayesian neural architecture that disentangles learning of the data distribution from inference during the learning process. It outperforms its counterparts based on point estimates.

**Strengths:**

It outperformed the VAE on two datasets of image classification.

**Weaknesses:**

Overall, this paper lacks novelty, motivation is unclear, experimentation is insufficient, and there is a lack of references concerning domain generalization.

### Novelty:

1. Using causality or Bayesian for domain generalization is not novel and already investigated in [1,2,3]. Specifically, [1,2] proposed causality inspired method to domain generalization. [3] proposed Bayesian learning to extract domain invariant features.

2. The proposed only works on image classification task and only CNN based architectures.

### Motivation:

1. The motivation is unclear. Could you elaborate on your rationale for using Bayesian Neural Networks for disentanglement? What are the benefits of combining domain-invariant and specific parts during inference?

### Experiments:

1. Adding ablation studies for KL divergence term of weights to demonstrate the necessary Bayesian network. Or only use domain-invariant part in the inference time.
2. Comparison methods are insufficient. Please compare all the method in the "Missing reference", and highlight the difference.
3. Only two simple datasets CIFAR10 and Office Home. Please add more datasets. e.g., DomainBed benchmark, which including five datasets (PACS, VLCS, OfficeHome, Terra, DomainNet) and is the most widely used benchmark for domain generalization.

4. Only CNN-based architecture. Please using other widely used backbone like Visual Transformer such as Vit-B-16.

5. Only test on image classification task, please test on time series task as well.

### Missing References:

[1] Lv, Fangrui, et al. "Causality inspired representation learning for domain generalization." Proceedings of the IEEE/CVF conference on computer vision and pattern recognition. 2022.

[2] Ouyang, Cheng, et al. "Causality-inspired single-source domain generalization for medical image segmentation." IEEE Transactions on Medical Imaging 42.4 (2022): 1095-1106.

[3] Xiao, Zehao, et al. "A bit more bayesian: Domain-invariant learning with uncertainty." International conference on machine learning. PMLR, 2021.

[4] Derakhshani, Mohammad Mahdi, et al. "Bayesian prompt learning for image-language model generalization." Proceedings of the IEEE/CVF International Conference on Computer Vision. 2023.

[5] Yu, Xi, et al. "INSURE: an Information theory iNspired diSentanglement and pURification modEl for domain generalization." IEEE Transactions on Image Processing (2024).

**Questions:**

1. Without a reconstruction loss, how can it ensure that no information is lost (like domain-invariant information) in achieving disentanglement?

2. Please add computation complexity in the table, since the objective function contains two KL terms.

3. How to choose the hyper-parameter in front of each terms?

---

> ### Author Response · Authors · 2024-11-23
>
> Thank you for your review, we are sorry that you find that our work lacks novelty and experiments and has unclear motivations. We hope that our explanations below can help clear out any confusions you may have about the paper. Regarding the lack of references, thank you very much for pointing it out, we acknowledge that our works lacks comparisons with other domain generalisation methods. We have integrated this feedback in our paper and added comparisons between our method and the existing, highlighting the conceptual differences between the methods and conducted comparative experiments on standard benchmarks.
>
>
> Experiments are currently running but we hope to add the results before the end of the discussion phase. In the meantime, please find below the answers to your other concerns:
>
> We would like to clarify in the strengths section that we compare out model with a base ResNet-18 corresponding to the backbone architecture to which we apply our method and not with a VAE. The second CT baseline contains a VAE as part of its architecture but it is also based on ResNet-18 as its backbone.
>
>
> Regarding your questions on the motivation of our work:
>
> 1. We aim to train a model that computes interventional queries P(y|do(x)). Our work is motivated by the observation that existing studies do not consider spurious correlations arising by the conditioning on the training set (as shown in Figure 1). From this observation, we derive a quantity that allows learning interventional queries while taking this bias into account. This derivation includes a marginalisation into the set of possible weights to block a backdoor path, justifying the use of Bayesian neural networks to perform this marginalisation. The rest of the architecture directly follow from this derivation. The combination of the input and contextual embeddings allows the model to differentiate domain-specific information (about the current distribution, provided by the context) and domain-invariant knowledge (learned from the input).
>
>
> Regarding your comments on the experiments:
>
> 1. We provide ablation studies where we modify the amount of sampling from BNN and inference component in Figures 5 and 6. These results show that using a BNN has little impact on the final performance but helps the learning to be more sample efficient.
>
> 2. & 3. We have added comparison with the missing references. Please, refer to the general comment above to see the details and additional experiments. We are currently running experiments on the datasets of the DomainBed benchmark.
>
> 4. & 5. Similarly to the domain generalisation baselines, our model is tailored for image classification tasks with CNNs. While our proposed method can theoretically be transferred to another backbone like the VIT and to other tasks, it would require many adaptations specific to these domain and architecture. Moreover, the domain generalization baselines are all based on ResNet, allowing comparison of the proposed methods for the same backbone model. Transferring our method to other backbones and domains is an interesting direction that we will consider as part of our future work.
>
>
> Regarding your additional questions:
>
> 1. Our theoretical work shows that a reconstruction loss is not needed for inference. Intuitively, this can be explained by the fact that a reconstruction loss incites the model to learn details that are domain-specific. We aim to learn domain-invariant functions by putting the domain information into the contextual image embeddings (from the marginalisation over the input domain).
>
> 2. We indeed use multiple KL divergence terms in our objective function but we only use them to regularise parametric distributions with the standard normal distribution. So the quantity can be simplified as follows: $KL(\mathcal{N}(\\nu,\sigma)||\mathcal{N}(0,1)) = \frac{1}{2} \sum\limits_{i=1}^D (\sigma_i^2 + \nu_i^2 -1 - \ln \sigma_i^2)$, which has a linear complexity with the dimension size D ($\mathcal{O}(D)$). Does this answer your question?
>
> 3. We run hyperparameter grid search on the validation set of CIFAR-10 to find optimal hyperparameter values.
>
> Please, let us know if our response answers your concerns and if you would consider raising your score in the light of additional experiments and baseline comparisons. Otherwise, let us know what else we should improve.

---

> > ### Comment · Reviewer_Ee74 · 2024-11-27
> >
> > Thank you for your response. It solves some of my questions. However, I remain unconvinced about the motivation for using BNN. Could you elaborate further on why BNN is particularly well-suited for learning interventional queries and why it is only applied in the inference component? Additionally, the experimental evaluation part is insufficient as it is limited to ResNet-18. Finally, the loss function includes five terms, requiring four hyperparameters to be tuned, which makes it challenging to generalize the method to different settings. Based on these concerns, I will maintain my current score.

---

> ### Author Response · Authors · 2024-12-04
>
> Thank you for your response and for your time. The inclusion of a BNN is motivated by the derivation of the interventional query. This query, by construction, removes the causal factors affecting the input $X$ and potential biases arising from spurious correlations or unbalanced distributions, thus representing a domain-invariant quantity. However, this quantity cannot be trivially estimated: our derivatoin choses that a frontdoor and a backdoor paths must be blocked to compute it from observations (by removing the do operation). The frontdoor path is bloked by the double marginalisation over $X$ and $R$ and the backdoor path is blocked by the third marginalisation over $W$ (using frontdoor and backdoor criteria). A BNN represents a distribution of weights instead of point-estimates and can be used to perform this marginalisation. This finding is supported by previous work that showed the effectiveness of BNNs to differentiate aleatoric and epistemic uncertainty [1,2]. We want the inference component to handle aleatoric/model uncertainty only and not be biased by the input distribution, thus we model it with a BNN. Following previous work on partially-stochastic BNNs [3], we only model part of the inference module to preserve efficiency.
>
> Regarding the experiments, we agree with your comment. We have duly noted the missing experiments and are currently working on them. We will include results on more benchmarks and models in the next version of the paper.
>
> Regarding the number of hyperparameters in the loss function, we conducted hyperparamer tuning experiments to evaluate the impact of the hyperparameter choice. We found that the three KL divergence terms are not correlated with performance for values between $10^{-5}$ and $10^{-7}$ and that the choice of the final hyperparameter has limited impact if set below $0.01$. these results show that these hyperparameters do not affect generalisation to different settings. We will include this analysis in the next version of the paper.
>
>
> [1] Kendall, A., & Gal, Y. (2017). What uncertainties do we need in bayesian deep learning for computer vision?. Advances in neural information processing systems, 30.
>
> [2] Magris, M., & Iosifidis, A. (2023). Bayesian learning for neural networks: an algorithmic survey. Artificial Intelligence Review, 56(10), 11773-11823.
>
> [3] Sharma, M., Farquhar, S., Nalisnick, E., & Rainforth, T. (2023, April). Do bayesian neural networks need to be fully stochastic?. In International Conference on Artificial Intelligence and Statistics (pp. 7694-7722). PMLR.

---

### Author Response · Authors · 2024-11-23

We thank all the reviewers for their extensive and constructive feedback. The reviewers agree that in its current stage, our paper does not provide sufficient comparison with other domain generalisation baselines. We are currently working on improving this part and we provide an in-depth comparison between our work and related studies on domain generalisation as suggested by the reviewers. We will add numerical comparison on standard benchmarks in a second general comment once the additional experiments are finished.

Regarding the conceptual differences between our method and existing work:

The Causal Semantic Generative (CSG) model [1] is a method for o.o.d generalisation relying on causal invariance. It differs from our work as it uses different assumptions on the causal generative mechanisms, i.e. CSG considers the input $X$ and output $Y$ to be caused by shared latent factors but not to be directly causally related, leading to a different derivation of the quantity to estimate. We aim to compute the interventional quantity P(Y|do(x)) while CSG optimises P(Y|$Z_R$). It means that CSG attempts to use domain-invariant information only while we take advantage of contextual domain-specific information (by learning to separate it from domain-invariant mechanisms). The method also assumes the prior knowledge on the distribution to be a factorisation of the domain-invariant and domain-specific variables, while we do not require this assumption. In addition, we explicitly model the domain shift using transportability theory and, as a result of the different derivation, we include marginalisations on the input domain and the inference weights to estimate interventional queries.

The Causality Inspired Representation Learning (CIRL) [2] is a similar method, taking advantage of a similar causal graph. The method considers that non-causal factors are important for classification and proposes to build contextual images by applying modifications of the input in the frequency domain (using Fourier transform). We similarly use contextual images to learn domain-specific information but we use sampling from the distribution to increase the diversity of the representation and not be conditioned on the instance label. CIRL encode input and contextual images simultaneously, similarly to our work, and uses a factorisation loss to induce the dimensions of the latent representations to represent causal factors and be independent. However, no guidance is provided as of how the causal factors should be disentangled (unsupervised learning settings), hereby providing no guarantees that the factors are disentangled or causal since it was shown by [3] that unsupervised disentanglement is generally impossible. Our work differs by learning the parameters of latent distributions instead of point estimates to regularise the latent space (both for the image embeddings and for the classifier weights). We also do not attempt to directly disentangle the causal factors (following the results of [3]) but only separate the causal and non-causal factors, relaxing CIRL assumptions.

[4] is a different approach to domain generalisation not relying on causal principles. Similarly to our work, the method learns the parameters of a domain-invariant distribution but does not take advantage of the domain-specific representation. The objective function also differs as the method does not aim to optimise an interventional query. Furthermore, in our work, we justify the use of Bayesian layers as part of the derivation of the quantity P(Y|dox(X)) to estimate (to block a backdoor path between the input domain and the output class).

---

> ### Author Response · Authors · 2024-11-23
>
> The Information theory iNspired diSentanglement and pURification modEl (INSURE) [5] is another method not relying on causality theory (although, information theory is arguably very closely linked [6]). It simultaneously learn a class representation and a domain representation, trained on domain and label classification tasks. A Mutual Information loss is used to incite the representations to be independent and a mixup strategy is used to further disentangle the two concepts by training the model on two images with inverted domain representations. Our work differs by learning the parameters of the latent distributions instead of point estimates to regularise the latent space and by representing domain-specific information implicitely. INSURE uses a domain prediction loss to disentangle class information from domain information while we use implicit regularisation to avoid overfitting to this auxiliary task and make sure this representation still contains useful information for class prediction, even out-of-distribution.
>
> [7,8,9,10,11] are studies applying causality theory and Bayesian methods to different problems than the one considered in our work: [7] uses causality to improve image segmentation on medical data and [8] uses a Bayesian method to learn model prompts for language models. [9,10,11] are causal reinforcement learning methods.
>
> We will include these explanations in the next version of the paper.
>
>
> [1] Liu, Chang, Xinwei Sun, Jindong Wang, Haoyue Tang, Tao Li, Tao Qin, Wei Chen, and Tie-Yan Liu. "Learning causal semantic representation for out-of-distribution prediction." NeurIPS, 2021.
>
> [2] Lv, Fangrui, et al. "Causality inspired representation learning for domain generalization." Proceedings of the IEEE/CVF conference on computer vision and pattern recognition. 2022.
>
> [3] Locatello, F., Bauer, S., Lucic, M., Raetsch, G., Gelly, S., Schölkopf, B., & Bachem, O. (2019, May). Challenging common assumptions in the unsupervised learning of disentangled representations. In international conference on machine learning (pp. 4114-4124). PMLR.
>
> [4] Xiao, Zehao, et al. "A bit more bayesian: Domain-invariant learning with uncertainty." International conference on machine learning. PMLR, 2021.
>
> [5] Yu, Xi, et al. "INSURE: an Information theory iNspired diSentanglement and pURification modEl for domain generalization." IEEE Transactions on Image Processing (2024).
>
> [6] Arjovsky, M., Bottou, L., Gulrajani, I., & Lopez-Paz, D. (2019). Invariant risk minimization. arXiv preprint arXiv:1907.02893.
>
> [7] Ouyang, Cheng, et al. "Causality-inspired single-source domain generalization for medical image segmentation." IEEE Transactions on Medical Imaging 42.4 (2022): 1095-1106.
>
> [8] Derakhshani, Mohammad Mahdi, et al. "Bayesian prompt learning for image-language model generalization." Proceedings of the IEEE/CVF International Conference on Computer Vision. 2023.
>
> [9] De Haan, Pim, Dinesh Jayaraman, and Sergey Levine. "Causal confusion in imitation learning." NeurIPS, 2029.
>
> [10] Zhang, Amy, Clare Lyle, Shagun Sodhani, Angelos Filos, Marta Kwiatkowska, Joelle Pineau, Yarin Gal, and Doina Precup. "Invariant causal prediction for block mdps." ICML, 2020.
>
> [11] Bica, Ioana, Daniel Jarrett, and Mihaela van der Schaar. "Invariant causal imitation learning for generalizable policies." NeurIPS, 2021.

---

> > ### Author Response · Authors · 2024-11-26
> >
> > As requested, we provide additional details on the differences between our work and studies [7,8,9,10,11] below:
> >
> > [7] augments input images using randomly initialised convolutional networks and trains a segmentation model to make the same prediction on the augmented version of the input image. While our approach differs as we use in-domain sampling, this augmentation approach is similar to that of [2], although this augmentation technique only modifies image intensity and texture and does not consider changes in shape or translations/rotations. This is well suited for medical images and for segmentation tasks as the augmentation does not alter the ground-truth segmentation mask, but not for the more general case considered in our work.
> >
> > [8] proposes a prompt learning method for image-language models. The aim of the paper is not to train a parametric model to solve a task but to optimise the input prompt given to a language model to boost its performance. To regularise training and reduce overfitting, the authors formulate the problem as estimating the prompt space distribution and use a Bayesian method to solve it. The problem settings are very different as the target distribution is a general prior over the possible prompts while we aim to compute an (interventional) posterior probability $P(Y|do(X))$.
> >
> > [9] tackles the problem of causal misidentification in imitation learning. The proposed method attempts to learn the true causal model behind the policy of experts acting in an environment. This problem differs from ours as it aims to discover the complete causal structure underlying a policy while we aim to elucidate one causal query given a known causal structure (see Figure 1). More importantly, this work assumes access to interventional data from the environment, either by querying an expert or by directly acting in the environment, while we are in a supervised settings and only have access to a fixed set of observations. [10] and [11] are similar reinforcement learning methods and share the same differences.

---

### Comment · Area_Chair_FVEK · 2024-11-24

Dear Reviewers,

This is a gentle reminder that the authors have submitted their rebuttal, and the discussion period will conclude on November 26th AoE. To ensure a constructive and meaningful discussion, we kindly ask that you review the rebuttal as soon as possible and verify if your questions and comments have been adequately addressed.

We greatly appreciate your time, effort, and thoughtful contributions to this process.

Best regards, AC

---

### Meta-Review · Area_Chair_FVEK · 2024-12-23

**Metareview:**

The paper proposes a theoretical framework combining causality principles with Bayesian neural networks. While the theoretical contribution and integration of recent advances in partially-stochastic Bayesian networks was appreciated, the paper has several critical limitations. The empirical validation is limited to just two datasets with only modest performance improvements, which don't convincingly demonstrate the practical value of the added model complexity. Finally, the paper does not justify architectural design decisions, limited tasks, and has insufficient comparisons against other recent domain generalization methods.

**Additional Comments On Reviewer Discussion:**

See above for salient concerns. The authors did not provide sufficient motivation and empirical evidence to justify the proposed method (reviewer Ee74) and establish clear differentiation with prior works (reviewer uvMu) in this space. These are the primary grounds for rejection of the work. The former especially is significant for the work to have long-term impact in this area.

---

### Decision · Program_Chairs · 2025-01-22

Reject